# *idpr*: A package for profiling and analyzing <u>I</u>ntrinsically <u>D</u>isordered <u>P</u>roteins in <u>R</u>

**William M. McFadden**[1¤], **Judith L. Yanowitz**[1,2]*

**1** Magee-Womens Research Institute, Pittsburgh, PA, United States of America, **2** Department of Obstetrics, Gynecology, and Reproductive Sciences, University of Pittsburgh School of Medicine, Pittsburgh, PA, United States of America

¤ Current address: Department of Pediatrics, Laboratory of Biochemical Pharmacology, Emory University School of Medicine, Atlanta, GA, United States of America

* yanowitzjl@mwri.magee.edu

**Data Availability Statement:** idpr is available for download at: https://doi.org/10.18129/B9.bioc.idpr.

**Funding:** This work was funded by National Institutes of Health, grant #R01GM127569 to Judith Yanowitz. https://public.era.nih.gov/

## Abstract

Intrinsically disordered proteins (IDPs) and intrinsically disordered regions (IDRs) are proteins or protein-domains that do not have a single native structure, rather, they are a class of flexible peptides that can rapidly adopt multiple conformations. IDPs are quite abundant, and their dynamic characteristics provide unique advantages for various biological processes. The field of "unstructured biology" has emerged, in part, because of numerous computational studies that had identified the unique characteristics of IDPs and IDRs. The package '*idpr*', short for **I**ntrinsically **D**isordered **P**roteins in **R**, implements several R functions that match the established characteristics of IDPs to protein sequences of interest. This includes calculations of residue composition, charge-hydropathy relationships, and predictions of intrinsic disorder. Additionally, *idpr* integrates several amino acid substitution matrices and calculators to supplement IDP-based workflows. Overall, *idpr* aims to integrate tools for the computational analysis of IDPs within R, facilitating the analysis of these important, yet under-characterized, proteins. The *idpr* package can be downloaded from Bioconductor (https://bioconductor.org/packages/idpr/).

## Introduction

Intrinsically disordered proteins (IDPs) are proteins that lack a single, rigid structure under native conditions [1–4], challenging the long-held paradigm that structure leads to function. In addition to typical cellular processes, IDPs have been implicated in human diseases such as neurodegenerative disorders and various cancers [5–7]. IDPs contain one or more intrinsically disordered region (IDR), which are regions of proteins composed of thirty or more disordered residues. Bioinformatic studies have shown that one-third to one-half of eukaryotic proteomes are predicted IDPs [8–11]. Further, viral proteomes appear to be enriched in IDPs, exemplified with the most disordered proteome observed belonging to the *Avian carcinoma virus* with an average disorder composition of over 77% [9].

Due to their apparent abundance and relevance, research interest in IDPs has been increasing [12]. In this regard, there are many computational tools that predict the intrinsic disorder

commonsplus/. The funders had and will not have a role in study design, data collection and analysis, decision to publish, or preparation of the manuscript.

**Competing interests:** he authors have declared that no competing interests exist.

within a protein sequence [13–15]. These tools utilize known differences between disordered and ordered proteins, such as the distinct compositional profile, evolutionary rate, and biochemical properties of IDPs and IDRs compared to proteins or protein-regions with compact, ordered structure [16–19]. Since IDPs have decreased levels of secondary and tertiary structures [1], the primary structure serves as the principal source of computational information for IDPs. Thus, most IDP prediction tools rely on the protein's sequence of amino acids, commonly represented as a character string of individual letters [13–15]. While several R packages analyze protein characteristics based on the amino acid sequence alone, to our knowledge, there is not been a package that is focused on the unique features of IDPs and IDRs.

The R package that we created borrows its acronym from "IDPR" or Intrinsically Disordered Protein Regions; *idpr* stands for "Intrinsically Disordered Proteins in R". The goal of this R package is to integrate tools for IDP analysis, including amino acid composition, charge, and hydropathy, using the R platform. Additional IDP analysis is facilitated by several amino acid substitution matrices that are IDP-specific [20–22] as well as linking to the suite of disorder predictions by IUPred2A [23, 24] retrieved by connection to their REST API. The *idpr* package can be found at https://bioconductor.org/packages/idpr/.

*idpr* aims to balance a workflow that automatically generates key visualizations for users of any skill level with a workflow that allows dynamic input and custom output for more-experienced users. The *ggplot2* package [25] is used to generate the visualizations, allowing users to access ggplot theme options and aesthetics for further customization. Additionally, *idpr* graphic functions give users the option to return calculations as values for downstream analysis. Overall, *idpr* aims to integrate multiple tools for the computational analysis of intrinsically disordered proteins within R.

## Methods

### A. Implementation

*idpr* is implemented as an open-source R [26] / Bioconductor [27] package under an LGPL-3 license. For integration with various packages, *idpr* functions accept protein sequences as character strings, vectors of individual amino acids, and XString objects from the Biostrings package [28]. Alternatively, functions can analyze sequences directly from.fasta files. Substitution matrices within this package can integrate with other R packages used for sequence analysis and multiple sequence alignments. Package dependencies include R version 4.1.3, *Biostrings* [28], *jsonlite* [29], and several tidverse packages [30] including *ggplot2* [25].

*idpr* is a well-documented package with detailed user manuals and function descriptions, generated with *roxygen2* [31]. This package also includes six vignette documents (long-form documentation) that discusses the theories of IDPs with the functionality of the package. Versions of *idpr* can be installed through the *BiocManager* package manager [32] from Bioconductor Release ≥3.13 (bioconductor.org) with the following line of R code: BiocManager::install ("idpr"). *idpr* version 1.5.11 was used for this publication and the workflow can be found in the supplementary materials or at dx.doi.org/10.17504/protocols.io.kqdg3p241l25/v1.

### B. 'idpprofile'

To quickly generate the *idpr* profile for a protein of interest, a UniProt ID and the amino acid sequence are used to create multiple plots with a single command. idpprofile() serves as a wrapping function for key graphing tools within *idpr*. These plots include: Charge-Hydropathy Plot, Local Charge Plot, Local Scaled Hydropathy Plot, Structural Tendency Plot, Compositional Profile Plot, and IUPred Plot, and FoldIndex Plot (Discussed Below). If a UniPot ID is

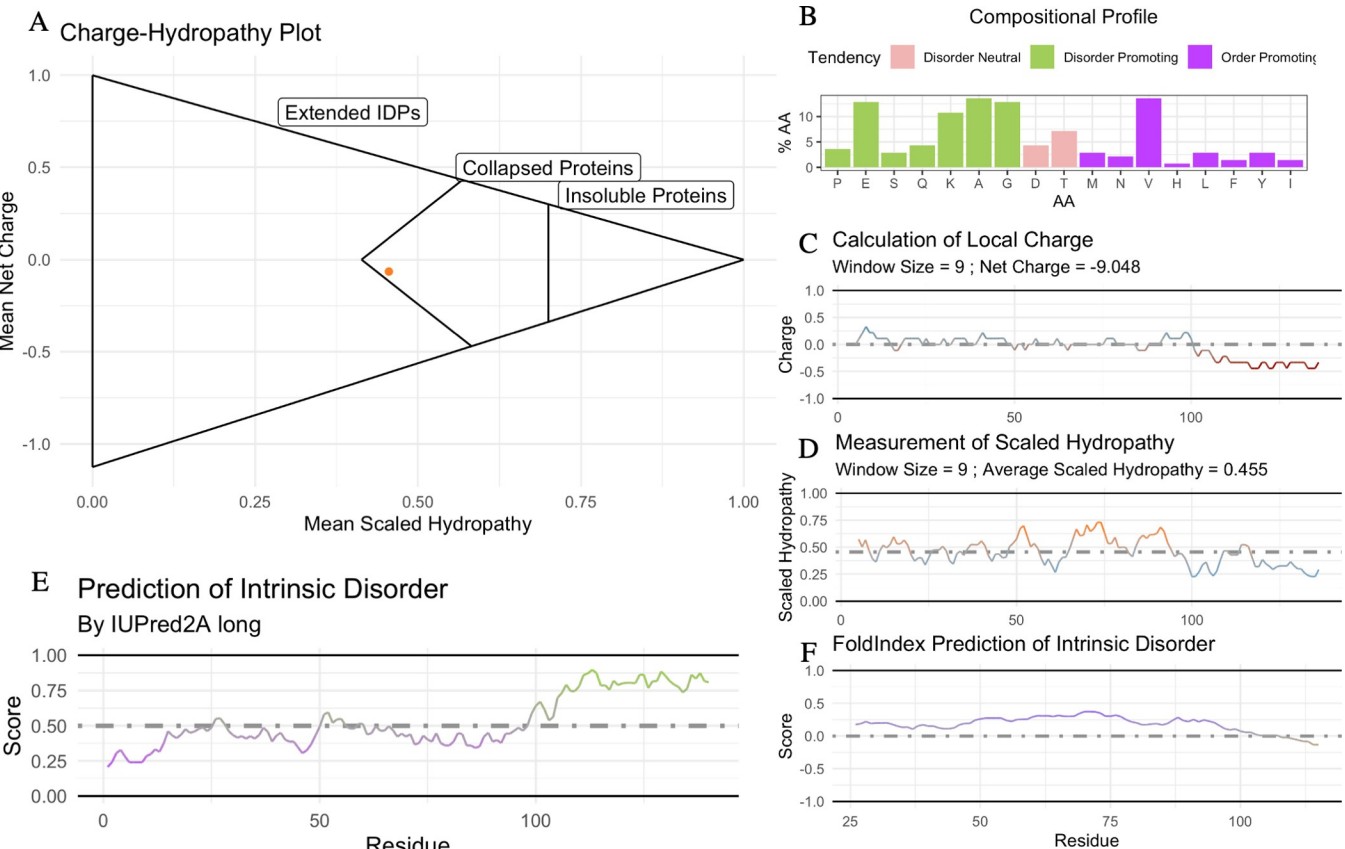

**Fig 1. The idprofile of α-Synuclein, generated by *idpr*, returns IDP characteristics.** (A) Charge-hydropathy plot of α-Synuclein (αSyn) predicts a collapsed protein. Method are described in [16]. Mean Scaled Hydropathy calculated with the Kyte and Doolittle measurement of hydropathy [33], scaled to Arg = 0.0 and Ile = 1.0. Mean Net Charge calculated with IPC_protein pKa values [34]. Cutoff equation is <Charge> = ±2.785<Hydropathy>±1.151 as described previously [18]. Proteins are considered insoluble when <Hydropathy> ≥ 0.7. (B) Structural tendency plot shows αSyn is enriched in disorder-promoting residues. Disorder-promoting residues (P, E, S, Q, K, A, and G) in green; order-promoting residues (M, N, V, H, L, F, Y, I, W, and C) in purple; disorder-neutral residues (D, T, and R) in pink [35]. (C) Local Charge Plot shows an acidic C-terminus. The local charge is the average of a 9 amino acid wide sliding window, calculated with the IPC_protein pKa values [34]. (D) Local Hydropathy Plot shows a C-terminus deficient in hydrophobic residues. The local hydropathy is the average of a 9 amino acid wide sliding window, calculated with the scaled Kyte and Doolittle measurement of hydropathy [33]. (E) IUPred2 predicts a C-terminal IDR in αSyn [23, 24]. Residues with a score 0.0–0.5 are predicted to be ordered, regions 0.5–1.0 are predicted to be disordered. (F) FoldIndex predicts a C-terminal IDR in αSyn [36]. Residues with a score 0.0 –+1.0 are predicted to be ordered, regions -1.0–0.0 are predicted to be disordered.

not included, the IUPred plot is skipped (Fig 1). Please refer to the supplementary workflow and package documentation for details on using idprofile and other *idpr* functions.

## C. Charge and hydropathy

It has been previously shown that both extreme net charge and deficiency in hydropathy are characteristics of intrinsic disorder proteins [16]. Extended IDPs will occupy a unique area on the plots of both average net charge and mean scaled hydropathy, [16]. meaning that the Charge-Hydropathy Plot can distinguish compact from extended proteins under native conditions (Fig 1A). One cannot, however, make a general rule about where IDPs on the spectrum from collapsed protein or an extended protein because IDPs can have the characteristics of either [16, 37].

Protein charges are calculated using the Henderson-Hasselbalch equation [38] with the IPC_protein pKa values [34] by default, although 15 additional pKa data sets are loaded into *idpr* for user preference. The Kyte and Doolittle measurement of hydropathy [33] are used,

scaled with Arg having a hydropathy of 0.0 and Ile having a hydropathy of 1.0. Local charges and local hydropathy are calculated using a sliding window to identify regions of interesting chemistry (Fig 1C and 1D). The sliding window is 9 residues by default but can be changed to any odd number. The resulting figure is similar to one that can be obtained by the ProtScale tool from ExPASy [39].

### D. Structural tendency

IDPs as a class tends to have a different composition of amino acids, and therefore distinct overall chemistry, from that of ordered proteins [40]. The chemistry of the specific residues influences its tendency to favor an extended or a compact structure. Residues enriched in the amino acid sequences of IDPs are typically charged, flexible, hydrophilic, or small; whereas order-promoting residues, found in structured proteins, tend to be hydrophobic, aromatic, aliphatic, or disulfide bond. There are also disorder-neutral residues [18, 35]. The default values, described previously [35], are disorder-promoting residues: P, E, S, Q, K, A, and G; order-promoting residues: M, N, V, H, L, F, Y, I, W, and C; and disorder-neutral residues: D, T, and R. These are represented by the structural tendency plot (Fig 1B). Other definitions of order- and disorder-promoting have been published [11], so users can opt to manually specify residue definitions.

### E. Disorder predictions

FoldIndex utilizes the described relationship of charge and hydropathy to identify unstructured regions of amino acid sequences [16, 36]. This method is implemented as part of many other prediction programs since it was described in 2005. Using a sliding window of size 51, a negative score ($<0$) indicates a region is predicted disordered; windows with a positive score ($>0$) are predicted as ordered [36]. Calculations are made with charge and hydropathy functions within *idpr* and uses IPC_protein pKa values [34] at pH 7.0 and the scaled Kyte and Doolittle measurement of hydropathy [33].

   The IUPred2 algorithm calculates a score of intrinsic disorder based on a model of the estimated energy potential for each residue interactions [23]. The structure in protein comes from a network of intramolecular interactions between amino acids. In IDPs, the (lack of) structure comes from the increase interactions of the amino acids with the surrounding environment. This reduced number of interactions leads to the IDP lacking secondary and tertiary structure [41]. IUPred2 predictions are made on a scale of 0.0–1.0, with 0.5 being the dividing line between order and disorder. $>0.5$ predicting a disordered region; $<0.5$ predicting an ordered region [23, 24, 41] (Fig 1E). An additional prediction of intermolecular protein-protein interactions is performed with the ANCHOR2 program (Fig 2A), and another predictor of redox-sensitive disorder is performed with IUPred2A Redox (Fig 2B) [23, 24, 41]. A Uniprot ID is required to access the IUPred2A REST API, as well as an internet connection. Visit the IUPred2A website (https://iupred2a.elte.hu/) for terms of use, references, and additional information.

### F. Visualizing discrete values

As mentioned above, specific amino acid residues are preferentially enriched in unstructured or ordered regions [35]. To visualize the location of assigned residue characteristics in the context of the amino acid sequence, *idpr* contains a way to visualize discrete values with a 'sequenceMap' (Fig 2C, S1B Fig in S1 File). This is not part of the idprofile function but is included within the package for additional investigation. The values visualized can be results from *idpr* or from any other source. This function can also visualize continuous values.

## G. Substitution matrices for analyzing IDPs

Because there is less restraint to maintain a specific 3D structure IDPs and IDRs tend to evolve faster than ordered proteins [17, 43]. Therefore, IDPs tend to accept increased point mutations at disparate rates when compared to ordered proteins [21].

Currently, PAM and BLOSUM are the most amino acid substitution matrices [44, 45], which are integrated into many web-based tools including NCBI-BLAST+ and EMBOSS [46, 47]. However, customization of the matrices is often desired but is not possible with these online programs. That said, BLOSUM and PAM matrices can both be used with alignment programs in R when loaded via the Biostrings Package [28]. However, for the analysis of IDPs, PAM and BLOSUM matrices are not ideals since they are derived from ordered proteins or favor residue substitutions common among structured proteins [20–22]. To circumvent this pitfall in *idpr*, EDSSMat [20], Disorder [21], and DUNMat [22] which provide IDP-derived substitution matrices have been incorporated for use in alignments.

## Results and discussion

### A. Example 1 - α-Synuclein

To highlight the use of the *idpr* package, α-Synuclein (αSyn; UniProt ID: P37840], is used in an example analysis. αSyn is an IDP, experimentally validated using various methods [48–52].

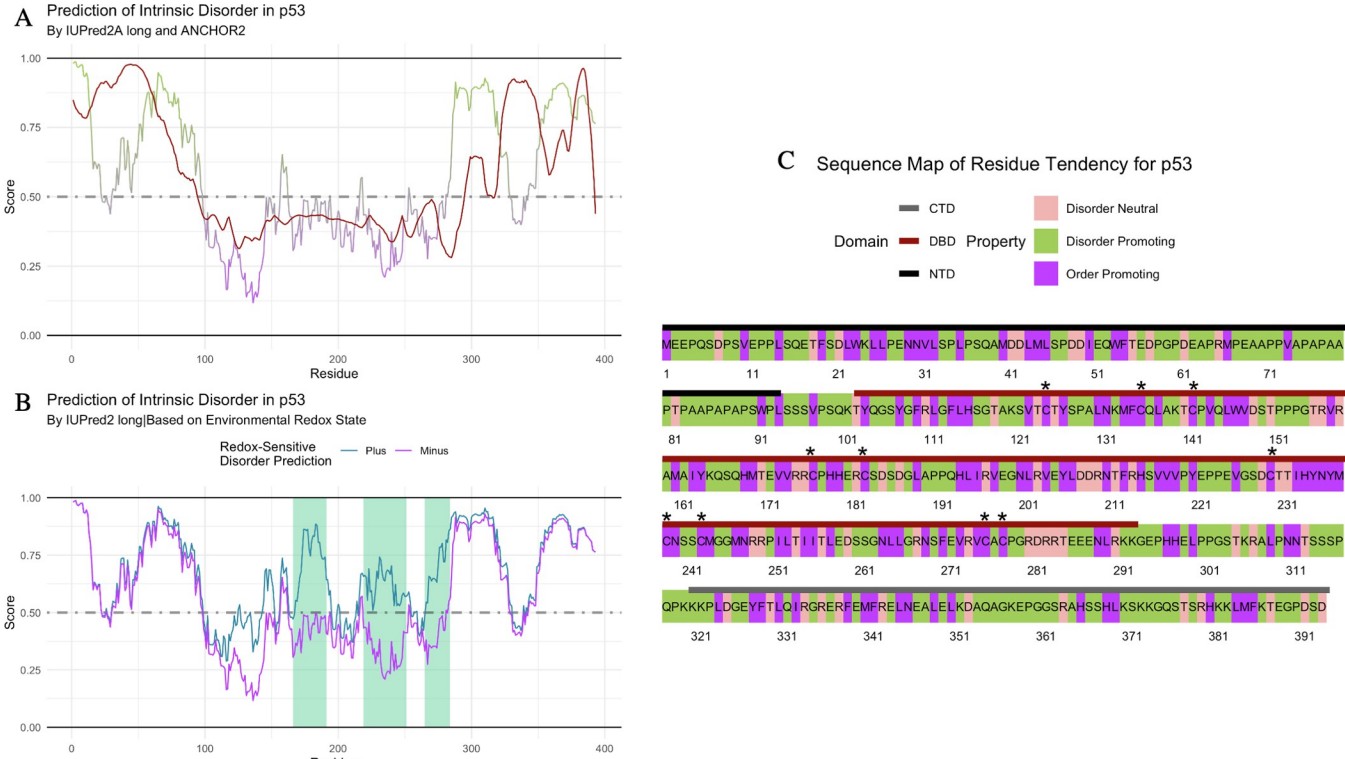

**Fig 2. Disorder predictions for p53 domains recapitulate environmental sensitivities.** (A) IUPred2A predicts multiple IDRs that promote protein-protein interactions in p53 [23, 24]. Residues with an IUPred2 score (green and purple line) of 0.0–0.5 are predicted to be ordered and residues 0.5–1.0 are predicted to be disordered. Residues with an ANCHOR2 score (red line) greater than 0.5 are predicted to be IDRs and protein-binding domains. (B) IUPred2A Redox predicts several oxidation-sensitive regions in p53 [23, 24]. Redox-plus (reducing environment) predictions are shown in blue, Redox-minus (oxidizing environment) predictions are shown in purple. Regions predicted as "Redox Sensitive" are highlighted in light green. Residues with an IUPred score of 0.0–0.5 are predicted to be ordered and residues 0.5–1.0 are predicted to be disordered. (C) Sequence map of structural tendency for each residue highlights the composition of p53 domains. N-terminal Domain (NTD) annotated by the black bar, DNA-Binding Domain (DBD) annotated by the red bar, C-terminal Domain (CTD) annotated by the grey bar. Conserved Cys residues (C124, C135, C141, C176, C182, C229, C238, C242, C275, C277] annotated [42]. Disorder-promoting residues (P, E, S, Q, K, A, and G) highlighted in green; order-promoting residues (M, N, V, H, L, F, Y, I, W, and C) in purple; disorder-neutral residues (D, T, and R) in pink [35].

This protein has been extensively studied and is heavily implicated in Parkinson's Disease pathology [53, 54]. The idprofile of αSyn returns IDP characteristics (Fig 1). The Charge-Hydropathy plot shows that αSyn appears to be a collapsed protein, rather than an extended IDP (Fig 1A). This is in line with previous reported data showing regions of αSyn are shielded from the cytoplasm under native conditions [49]. The structural tendency plot shows that αSyn is enriched in disorder-promoting residues, mostly represented by Glu, Lys, Ala, and Gly (Fig 1B). Interestingly, αSyn lacks Cys and Trp, both of which are the most order-promoting residues [35], in addition to lacking Arg, a positively charged and order-neutral residue. The local charges of the protein are mostly neutral, apart from a negatively charged C-terminal region (Fig 1C). In conjunction with the local charge, the C-terminal region is deficient in hydrophobic residues, as shown by the local scaled hydropathy (Fig 1D). In fact, it has been reported that residues 104–140 of αSyn are more extended than the N-terminal portion of the protein [51]. The Charge-Hydropathy plot of αSyn residues 104–140 returns an extended IDR, while residues 1–103 returns a collapsed protein with a more neutral charge (S1A Fig in S1 File). This is in line with the IUPred2 and FoldIndex predictions of intrinsic disorder for αSyn, which shows the C-terminal region predicted as disordered (Fig 1E and 1F, S1B Fig in S1 File). There are known point mutations in αSyn that are associated with familial Parkinson's Disease: A30P, E46K, H50Q, G51D, and A53T [55–57]. While these mutations are located in the more compact region of the protein, most mutations occur in disorder-promoting residues, with the exception of H50Q (S1B Fig in S1 File). Overall, the idprofile is useful for identifying biochemical features related to IDRs within a protein of interest.

## B. Example 2—p53

Another well characterized IDP is the cellular tumor antigen p53 (UniProt ID: P04637]. p53 has been studied extensively since it is mutated in over 50% of human cancers [58, 59]. It is an experimentally validated IDP [60–62] that acts as a protein hub, interacting with many different partners [3, 63]. The idprofile of p53 shows characteristics of a protein with several IDRs (S2 Fig in S1 File). The C-terminal domain (CTD) of p53 has been highly studied due to its ability to reversibly form various secondary structures depending on the specific binding partner studied [3, 60, 63]. For example, residues 377–388 gain an α-helical structure when interacting with S100 calcium-binding protein B, while in the same region, residues 379–387, form a β-strand when interacting with Sirtuin [3, 60, 63]. To predict such regions, ANCHOR2 scores, produced by IUPred2A, predict domains that are disordered and are protein-binding regions which may undergo a gain-of-structure when bound [23, 24]. For p53, ANCHOR2 predicts binding in multiple IDRs, including the CTD and recapitulates the known disorder-to-order transition of this domain mentioned above (Fig 2A).

There are several evolutionarily conserved cysteines within p53, most of which are within the central DNA-binding domain (DBD) [42, 64]. Further, p53 has reported roles in redox regulation [42, 65, 66]. To this point, IUPred2 contains a context-dependent predictor of disorder distinguishing between reducing (plus) or oxidizing (minus) environments that can be used to predict redox-sensitive IDRs that may experience induced folding [23, 24]. IUPred2 Redox predicts that p53 has multiple regions of redox sensitivity in the DBD (Fig 2B). This is in line with the known impact of redox conditions on the DBD that influences the structure—and the subsequent function—of p53, consistent with published literature [64, 65, 67, 68]. The structural tendency of each residue in p53 is highlighted in a sequence map with domains [60] and conserved Cys residues [42] annotated (Fig 2C). This p53 analysis exemplifies the use of functions within *idpr* that are not automatically generated using the idprofile wrapping function.

## C. Example 3 –mouse GCNA

The germ cell nuclear acidic protein (GCNA) is required for male fertility and has roles in repairing DNA-protein crosslinks [69–71]. GCNA has orthologs from single-celled protists to mammals. In most species, the N-terminal half of GCNA is disordered and the C-terminal half contains an Sprt-Like metalloprotease domain, zinc finger, and HMG box. While there appears to be occasional losses of either the protease, HMG box, or zinc finger, all GCNA orthologs contain the IDR [70]. The IDR of GCNA is enriched in acidic residues, which contributes to the disordered nature. Interestingly, the mouse GCNA lacks all of the structured domains and was previously predicted to be entirely disordered by IUPRED [70, 72].

The idpprofile of mouse GCNA (UniProt ID: A0A1D9BZF0) displays that of an unstructured protein (Fig 3A). There is a long stretch of acidic residues, with glutamic acid (E) being the most abundant residue in the amino acid sequence (Fig 3B and 3C). Further, there is a significant enrichment of disorder promoting residues in mouse GCNA (Fig 3B), aligning with the previously reported amino acid composition of GCNA being similar to that of Disprot, a

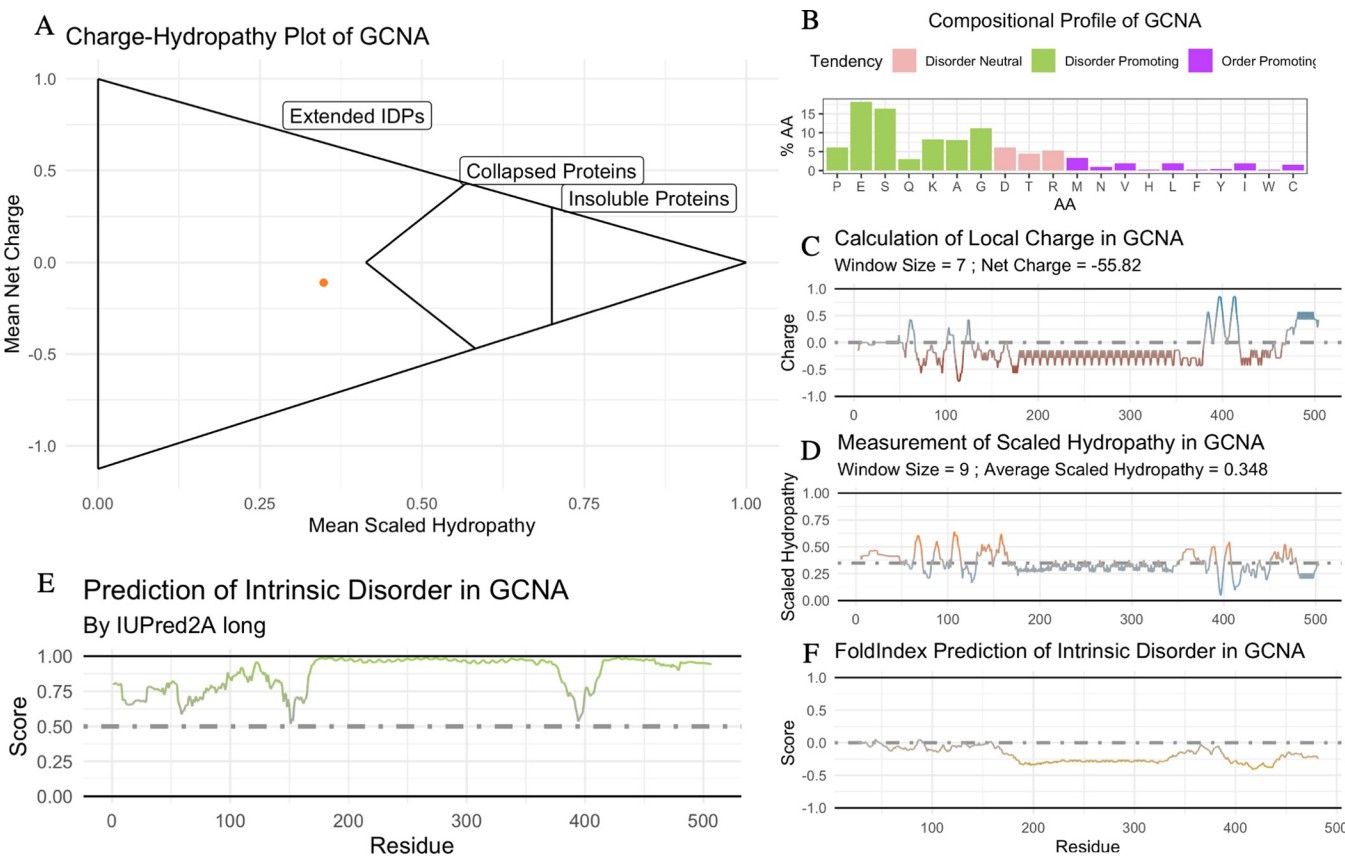

**Fig 3. The idpprofile of mouse GCNA shows prediction of an entirely disordered IDP.** (A) Charge-hydropathy plot of GCNA predicts a disordered protein. Method are described in [16]. Mean Scaled Hydropathy calculated with the Kyte and Doolittle measurement of hydropathy [33], scaled to Arg = 0.0 and Ile = 1.0. Mean Net Charge calculated with IPC_protein pKa values [34]. Cutoff equation is $<Charge> = \pm2.785<Hydropathy>\pm1.151$ as described previously [18]. Proteins are considered insoluble when $<Hydropathy> \geq 0.7$. (B) Structural tendency plot shows GCNA is enriched in disorder-promoting residues. Disorder-promoting residues (P, E, S, Q, K, A, and G) in green; order-promoting residues (M, N, V, H, L, F, Y, I, W, and C) in purple; disorder-neutral residues (D, T, and R) in pink [35]. (C) Local Charge Plot shows an acidic C-terminus. The local charge is the average of a 7 amino acid wide sliding window, calculated with the IPC_protein pKa values [34]. (D) Local Hydropathy Plot shows a C-terminus deficient in hydrophobic residues. The local hydropathy is the average of a 9 amino acid wide sliding window, calculated with the scaled Kyte and Doolittle measurement of hydropathy [33]. (E) IUPred2 predicts a C-terminal IDR in GCNA [23, 24]. Residues with a score 0.0–0.5 are predicted to be ordered, regions 0.5–1.0 are predicted to be disordered. (F) FoldIndex predicts a C-terminal IDR in GCNA [36]. Residues with a score 0.0 –+1.0 are predicted to be ordered, regions -1.0–0.0 are predicted to be disordered.

database of intrinsically disordered proteins [70, 73]. There are very few hydrophobic residues, and the protein has an average scaled hydropathy of 0.348 (Fig 3D). Both the extreme acidity of GCNA and the enrichment of disorder-promoting soluble residues contribute to the entire peptide being predicted as disordered from N- to C-terminus (Fig 3E and 3F). This replicates previously reported predictions of mouse GCNA being disordered [70].

## Conclusion

We have created an integrated R package that combines disorder prediction tools, hydropathy, and amino acid composition to facilitate the characterization of IDPs. The presence of charge repulsion and hydrophobic deficiencies are hallmark characteristics of an IDP or IDR [18]. The *idpr* package contains distinct, customizable methods for calculating charge and hydropathy for a protein sequence of interest. The output is a visually accessible, graphical readout of critical parameter for IDP analysis. We have validated the use of this tool with α-Synuclein, p53, and GCNA.

A significant portion of the eukaryotic proteome is thought to contain IDRs, but our understanding of these domains is still lacking. In some cases, these domains serve as bridges between two structured domains [74]. In others, like p53, the IDR attains different structure with unique protein partners [3, 75]. Yet in others, the IDRs support liquid-liquid phase separation [76]. In most cases, the role of the IDR is unknown. By providing an integrative tool for characterization of these domains, we envision *idpr* as platform upon which to find commonalities between IDPs and all for sub-division of these protein families.

## Supporting information

**S1 File. This contains a list of abbreviations and S1 and S2 Figs.**
(DOCX)

**S2 File. The code used to generate all graphics presented in this manuscript.**
(PDF)

## Acknowledgments

We would like to acknowledge the support of Dr. Michael Buszczak during the development of the package. Additionally, we would like to thank Dr. Miguel Brieño-Enríquez and the members of the Brieño-Enríquez and Yanowitz labs for feedback during package development. For providing essential details in making the package, we would like to acknowledge the book *R Packages* by Hadley Wickham (O'Reilly) ©2015 Hadley Wickham, ISBN: 978-1-491-91059-7.

## Author Contributions

**Conceptualization:** William M. McFadden.

**Funding acquisition:** Judith L. Yanowitz.

**Methodology:** William M. McFadden.

**Project administration:** Judith L. Yanowitz.

**Software:** William M. McFadden.

**Supervision:** Judith L. Yanowitz.

**Visualization:** William M. McFadden.

**Writing – review & editing:** Judith L. Yanowitz.

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
