## [Decision Letter · Decision Letter 0]

7 Jan 2022

PONE-D-21-38609idpr: A package for profiling and analyzing Intrinsically Disordered Proteins in RPLOS ONE

Dear Dr. Yanowitz,

Thank you for submitting your manuscript to PLOS ONE. After careful consideration, we feel that it has merit but does not fully meet PLOS ONE’s publication criteria as it currently stands. Therefore, we invite you to submit a revised version of the manuscript that addresses the points raised during the review process.

ACADEMIC EDITOR:Please try to improve your manuscript according to the reviewers' criticism. It would be good to add some more examples of successful predictions carriied out by your software package. 

We look forward to receiving your revised manuscript.

Kind regards,

Eugene A. Permyakov, Ph.D., Dr.Sci.

Academic Editor

PLOS ONE

Journal Requirements:

2. To comply with PLOS ONE submissions requirements, please provide the Protocols.io DOI in the Methods section of the manuscript using this format: “The protocol described in this peer-reviewed article is published on protocols.io, https://dx.doi.org/10.17504/protocols.io[........] and is included for printing as supporting information file 1 with this article.” Please also provide the Protocols.io DOI in the “Protocol DOI” field of the submission form (via “Edit Submission”). For more information, please see our submission guidelines:  https://journals.plos.org/plosone/s/submission-guidelines#loc-guidelines-for-specific-study-types.

3. We note you have not provided a Protocol.io PDF version of your protocol. As noted in our submission requirements, please upload a Protocol.io PDF version of your protocol as a Supporting Information file and name the file ‘S1 file’. Please update your Supporting Information Captions if necessary. If you have not yet uploaded your protocol to Protocols.io you are welcome to use the Protocols.io customer service code ‘PLOS2021.’ When using this customer code while submitting to Protocols.io, please make reference to your PLOS ONE submission, including your PLOS ONE manuscript number. With this customer code, Protocols.io editorial staff will import and format your protocol at no charge. For more information, please see our submission guidelines:  https://journals.plos.org/plosone/s/submission-guidelines#loc-guidelines-for-specific-study-types.

5. We noticed you have some minor occurrence of overlapping text with the following previous publication(s), which needs to be addressed:

- https://rdrr.io/bioc/idpr/f/inst/doc/idpr-vignette.Rmd

In your revision ensure you cite all your sources (including your own works), and quote or rephrase any duplicated text outside the methods section. Further consideration is dependent on these concerns being addressed.

Reviewers' comments:

Reviewer's Responses to Questions

**Comments to the Author**

1. Does the manuscript report a protocol which is of utility to the research community and adds value to the published literature?

Reviewer #1: Yes

Reviewer #2: Yes

2. Has the protocol been described in sufficient detail?

Descriptions of methods and reagents contained in the step-by-step protocol should be reported in sufficient detail for another researcher to reproduce all experiments and analyses. The protocol should describe the appropriate controls, sample sizes and replication needed to ensure that the data are robust and reproducible.

Reviewer #1: Yes

Reviewer #2: Yes

3. Does the protocol describe a validated method?

Reviewer #1: No

Reviewer #2: Yes

4. If the manuscript contains new data, have the authors made this data fully available?

Reviewer #1: Yes

Reviewer #2: N/A

**5. Is the article presented in an intelligible fashion and written in standard English?**

Reviewer #1: Yes

Reviewer #2: Yes

6. Review Comments to the Author

Reviewer #1: The manuscript PONE-D-21-38609 presents “idpr”, a software package aimed at the analysis of sequence properties of Intrinsically Disordered Proteins (IDPs) and unfolded protein regions, written by using the programming language R (one of the most common for statistical computing, data mining, and graphical representation). Following the general practice/tradition of reports of software packages, the manuscript illustrates the main features of “idpr” in a succinct way, includes test cases for two well-known IDPs (alpha-synuclein and p53), and refers to the program at the internet address https://bioconductor.org/packages/idpr/ for further info; a workflow including instructions for the installation is also provided as Supporting Information.

This work is interesting and provides a useful tool, because IDPs are currently widely investigated - as explained in the text, they are increasingly recognized as ubiquitously present in all organisms, and heavily involved in many pathologies. The language used is concise and direct, and the presentation is clear enough. Overall, it can be improved as described hereafter.

Major points:

- The two test cases presented both involve IDPs that possess large ordered regions. Alpha-synuclein in its functional conformation has 66% of its sequence in helical conformation, and such secondary structure is even folded in a sort of tertiary hairpin structure. Similarly, p53 has a high structural plasticity, but 63% of the sequence adopts a (labile) secondary and tertiary structure. A test case for an entirely unfolded IDP is lacking, and must be added to fully validate the software/protocol presented.

Any full-unstructured IDP can be chosen to this aim. For instance, NUPR1 (UniProt ID: O60356), a small protein of 82 residues with 0% secondary/tertiary structure (in spite of the wrong prediction of AlphaFold shown on the UniProt page). This protein remains disordered even in molecular complexes, thus is considered a model for a “perfect” IDP. Peaks in the hydropathy plot of NUPR1 nicely identify hot spots for the binding to molecular partners that include other (folded/unfolded) proteins, peptides, DNA, inorganic polymers, and various drugs, thus it constitutes an interesting and very easy test case. Other proteins, on top of my head, may include p21, prothymosin alpha, and perhaps even some unstructured proteins of SARS-CoV-1/2.

Minor points:

- Abstract: please consider mentioning https://bioconductor.org/packages/idpr/ already there or, alternatively, as soon as possible.

- Line 27: “these proteins have been implicated in several human diseases such as Parkinson’s Disease, Alzheimer’s Disease, and various cancers (5-7)”. Given the enormous progress of the research in this field, it is disappointing to see that the only papers cited are 10-15 years old. Please consider substituting or complementing them with updated references.

- Line 46: “the R package idpr stands for a few things: “Intrinsically Disordered Proteins in R” and “IDp PRofiles””. This seems to contradict the Abstract, where a single meaning of the acronym is given. Please correct either the Abstract or, more easily, the text, e.g. “stands for “Intrinsically Disordered Proteins in R”, although other acronyms such as “IDp PRofiles” are possible”.

- Line 64: “fasta files”. Please call it either FASTA (the correct name for the format) or .fasta (its file extension).

- Line 67: “tidverse packages”. Typo, it is “tidyverse”.

- Line 77: “If a UniPot ID is not included, the IUPred plot is skipped”. What happens if one wants to investigate a sequence that has not a UniProt ID, e.g. a new mutant of a known sequence? If this is already possible, it should be explained in the text; otherwise, it should be addressed in a future release of “idpr”.

- Line 81: “Method described in (16). Mean Scaled Hydropathy calculated with The Kyte and Doolittle measurement of hydropathy”. It should be “Methods are described...”, and “the Kyte and Doolittle” (lowercase “the”).

- Line 98: “extreme net charge and deficiency in hydropathy are characteristics of intrinsic disorder (16)”. It should be specified “intrinsic disorder in proteins” (or polypeptides).

- Line 109: “ The resulting figure is similar to ProtScale from ExPASy (38)”. It should be a bit expanded, e.g. “similar to the one that can be obtained by using the ProtScale tool from ExPASy”.

- Line 116: “Order promoting residues, meaning those enriched in structured proteins, tend to be aliphatic, hydrophobic, aromatic, or can form tertiary structures”. It should be “those more frequent in structured proteins” (structured proteins are enriched of these residues, not the other way around). Also, it should be “prone to form secondary/tertiary structures” (any residue “can form” such structures, with a few exception such as Prolines in secondary structures; the difference is again in the frequency).

- Line 117: “Disorder neutral residues”. It should be “Disorder-neutral residues”, the hyphen is crucial. Please also delete the space in “disorder- neutral residues”, three lines below.

- Line 139: “Residues with an IUPred2 long score”. The term “long” is unclear.

- Line 162: “This function can also visualize continuous values”. I cannot understand why it is important to specify this, or imagine an example of an use of such continuous values.

- Line 181: “aSyn is an IDP, experimentally determined using various methods”. Please use “investigated”, “validated”, “identified”, etc., instead of “determined”.

- Line 189: “Although the protein is deficient in Arg, the local charges of the protein are mostly neutral, apart from...”. I do not see the point of using “Although”.

- Line 201: “identifying biochemical features related to IDPs within a protein of interest”. Either it was meant “IDRs” instead of “IDPs”, or it is not clear.

- Discussion. This looks more like a Conclusion to me; maybe it can be slightly enlarged.

- Acknowledgments: It would be nice to state explicitly the names of Dr. M. Buszczak and Dr. M. Brieno-Enriquez.

Reviewer #2: The manuscript describes a relative simple but useful tool for profiling and analysis of intrinsically disordered proteins. The tool is available in R and provides quick analysis and visualization of several relative basic sequence properties of IDPs. These tools should be useful for someone who needs to do quick analysis of these sequence based properties. However, I have some reservation on how useful this tool will be. These properties are quite basic and the targeted users are probably non-experts; yet it require some proficiency in R and scripting to use. It would appear to be much more useful to have a web-based interface for the targeted users. I also recommend the author to build interface to other IDP analysis tools besides IUPred, such as Rohit Pappu's CIDER and others.

7. PLOS authors have the option to publish the peer review history of their article (what does this mean?). If published, this will include your full peer review and any attached files.

Reviewer #1: No

Reviewer #2: No

---

## [Author Response · Author response to Decision Letter 0]

28 Mar 2022

Response to Reviewers

We thank both reviewers for their time and thoughtful comments. They have helped to improve the manuscript and the functionality of the tool itself. Please find below our detailed responses to each point.

Reviewer #1: The manuscript PONE-D-21-38609 presents “idpr”, a software package aimed at the analysis of sequence properties of Intrinsically Disordered Proteins (IDPs) and unfolded protein regions, written by using the programming language R (one of the most common for statistical computing, data mining, and graphical representation). Following the general practice/tradition of reports of software packages, the manuscript illustrates the main features of “idpr” in a succinct way, includes test cases for two well-known IDPs (alpha-synuclein and p53), and refers to the program at the internet address https://bioconductor.org/packages/idpr/ for further info; a workflow including instructions for the installation is also provided as Supporting Information.

This work is interesting and provides a useful tool, because IDPs are currently widely investigated - as explained in the text, they are increasingly recognized as ubiquitously present in all organisms, and heavily involved in many pathologies. The language used is concise and direct, and the presentation is clear enough. Overall, it can be improved as described hereafter.

Major points:

- The two test cases presented both involve IDPs that possess large ordered regions. Alpha-synuclein in its functional conformation has 66% of its sequence in helical conformation, and such secondary structure is even folded in a sort of tertiary hairpin structure. Similarly, p53 has a high structural plasticity, but 63% of the sequence adopts a (labile) secondary and tertiary structure. A test case for an entirely unfolded IDP is lacking, and must be added to fully validate the software/protocol presented.

Any full-unstructured IDP can be chosen to this aim. For instance, NUPR1 (UniProt ID: O60356), a small protein of 82 residues with 0% secondary/tertiary structure (in spite of the wrong prediction of AlphaFold shown on the UniProt page). This protein remains disordered even in molecular complexes, thus is considered a model for a “perfect” IDP. Peaks in the hydropathy plot of NUPR1 nicely identify hot spots for the binding to molecular partners that include other (folded/unfolded) proteins, peptides, DNA, inorganic polymers, and various drugs, thus it constitutes an interesting and very easy test case. Other proteins, on top of my head, may include p21, prothymosin alpha, and perhaps even some unstructured proteins of SARS-CoV-1/2.

We thank the reviewer for their thoughtful comments and their consideration of our manuscript. Your main critique is an excellent point, thank you for this suggestion. We have added a section describing a protein that our lab is currently investigating. As described in the updated manuscript, the Germ Cell Nuclear Acidic protein (GCNA) from Mus musculus is previously reported as, and is predicted to be, entirely disordered. This protein also showcases the relationship of charge and hydropathy previously discussed in the manuscript. 

Minor points:

- Abstract: please consider mentioning https://bioconductor.org/packages/idpr/already there or, alternatively, as soon as possible. 

We have added the URL to the abstract.

- Line 27: “these proteins have been implicated in several human diseases such as Parkinson’s Disease, Alzheimer’s Disease, and various cancers (5-7)”. Given the enormous progress of the research in this field, it is disappointing to see that the only papers cited are 10-15 years old. Please consider substituting or complementing them with updated references.

We have added reference Zbinden, A., Pérez-Berlanga, M., De Rossi, P., and Polymenidou, M. (2020) Phase Separation and Neurodegenerative Diseases: A Disturbance in the Force, Developmental Cell 55, 45-68.

- Line 46: “the R package idpr stands for a few things: “Intrinsically Disordered Proteins in R” and “IDp PRofiles””. This seems to contradict the Abstract, where a single meaning of the acronym is given. Please correct either the Abstract or, more easily, the text, e.g. “stands for “Intrinsically Disordered Proteins in R”, although other acronyms such as “IDp PRofiles” are possible”. Addressed by removing multiple acronyms. 

- Line 64: “fasta files”. Please call it either FASTA (the correct name for the format) or .fasta (its file extension). Addressed.

- Line 67: “tidverse packages”. Typo, it is “tidyverse”. 

This is not a typo as some idpr package dependencies are: ggplot2 (>= 3.3.0), magrittr (>= 1.5), dplyr (>= 0.8.5), plyr (>= 1.8.6), which are tidyverse packages https://www.tidyverse.org/packages/ . idpr does not require the user to have the entirety of tidyverse installed. 

- Line 77: “If a UniPot ID is not included, the IUPred plot is skipped”. What happens if one wants to investigate a sequence that has not a UniProt ID, e.g. a new mutant of a known sequence? If this is already possible, it should be explained in the text; otherwise, it should be addressed in a future release of “idpr”.

The IUPred2A web interface allows users to submit any custom sequence, however this cannot be done via the R programming language. The REST API on the IUPred2A server requires a UniProt ID to fetch any prediction, thus we are limited in predictions. Within the idpr package, the iupred(), iupredAnchor(), and iupredRedox() functions’ help pages and the idpr user manual contains a URL to direct users to the IUPred2A website. Additionally, during revisions we have implemented a second method to predict intrinsically disordered regions, FoldIndex, which does not require a UniProt ID and predicts disorder within R. 

- Line 81: “Method described in (16). Mean Scaled Hydropathy calculated with The Kyte and Doolittle measurement of hydropathy”. It should be “Methods are described...”, and “the Kyte and Doolittle” (lowercase “the”). 

 Addressed.

- Line 98: “extreme net charge and deficiency in hydropathy are characteristics of intrinsic disorder (16)”. It should be specified “intrinsic disorder in proteins” (or polypeptides).

Addressed.

- Line 109: “ The resulting figure is similar to ProtScale from ExPASy (38)”. It should be a bit expanded, e.g. “similar to the one that can be obtained by using the ProtScale tool from ExPASy”.

Addressed.

- Line 116: “Order promoting residues, meaning those enriched in structured proteins, tend to be aliphatic, hydrophobic, aromatic, or can form tertiary structures”. It should be “those more frequent in structured proteins” (structured proteins are enriched of these residues, not the other way around). Also, it should be “prone to form secondary/tertiary structures” (any residue “can form” such structures, with a few exception such as Prolines in secondary structures; the difference is again in the frequency).

Addressed by adding in “sequences of structured proteins”. Addressed by changing “tertiary structures” to “disulfide bonds”. 

- Line 117: “Disorder neutral residues”. It should be “Disorder-neutral residues”, the hyphen is crucial. Please also delete the space in “disorder- neutral residues”, three lines below.

Addressed.

- Line 139: “Residues with an IUPred2 long score”. The term “long” is unclear.

Addressed by removing ”long”. 

- Line 162: “This function can also visualize continuous values”. I cannot understand why it is important to specify this, or imagine an example of an use of such continuous values.

This is based on the arguments of the ggplot2 package aesthetics that require different color pallets and theme arguments if discrete or continuous variables are visualized. Compare the sequenceMaps in figure 2C which visualizes discrete labels of amino acids to Figure S1B which visualizes continuous values of IUPred2 prediction of intrinsic disorder. 

- Line 181: “aSyn is an IDP, experimentally determined using various methods”. Please use “investigated”, “validated”, “identified”, etc., instead of “determined”. Addressed.

- Line 189: “Although the protein is deficient in Arg, the local charges of the protein are mostly neutral, apart from...”. I do not see the point of using “Although”. Addressed.

- Line 201: “identifying biochemical features related to IDPs within a protein of interest”. Either it was meant “IDRs” instead of “IDPs”, or it is not clear. Addressed.

- Discussion. This looks more like a Conclusion to me; maybe it can be slightly enlarged.

We have modified the manuscript headings to reflect this change as we agree this section represents our conclusion. Along with the resulting plots for each of the test cases presented, our results section includes discussion of how previous reports are reflected in these results. Thus, we have changed the “Results” section to “Results and Discussion” and changed the “Discussion” section to “Conclusions”.

- Acknowledgments: It would be nice to state explicitly the names of Dr. M. Buszczak and Dr. M. Brieno-Enriquez. 

Addressed.

Reviewer #2: The manuscript describes a relative simple but useful tool for profiling and analysis of intrinsically disordered proteins. The tool is available in R and provides quick analysis and visualization of several relative basic sequence properties of IDPs. These tools should be useful for someone who needs to do quick analysis of these sequence based properties. However, I have some reservation on how useful this tool will be. These properties are quite basic and the targeted users are probably non-experts; yet it require some proficiency in R and scripting to use. It would appear to be much more useful to have a web-based interface for the targeted users. I also recommend the author to build interface to other IDP analysis tools besides IUPred, such as Rohit Pappu's CIDER and others.

We thank the reviewer for their comments and consideration of our manuscript. We have added another prediction method to the idpr package, FoldIndex. This is now publicly available, documented within the idpr package, and has been updated throughout the manuscript. We thank you for suggesting that we add another prediction method, as this strengthens the utility of the package. 

The goal of the idpr is to simplify these calculations for R users of any skill level. We thank you for raising concern and we have tried to clarify this in the manuscript. Thus, the package has extensive documentation, with a 45-page reference manual and six additional Vignettes (R’s long-form documentation), to guide inexperienced users. To simplify the use for quick analysis, the idprofile serves as a wrapping function for key methods within the package and can return 6 plots with a single line code so long as the user has the sequence .fasta file and the idpr package downloaded: 

```{r}

idpr::idprofile(sequence = “path.fasta”, uniprotAccession = “ID”) 

```

However, experienced users will not require this wrapping function and have many options available besides the default settings. All graphing functions in idpr can be set to return results as a data frame containing the amino acid sequence in one column and calculated values in the second column. This enables users to perform downstream analysis, run statistical tests, or create their own custom plots. Thus, idpr is both a calculator for IDP characteristics and a tool to quickly visualize these values for analysis within R. Many prediction tools on the web exist for users who do not have coding experience, what is novel about this package is that it is the first for the R programming language.

---

## [Editor Report · Decision Letter 1]

30 Mar 2022

idpr: A package for profiling and analyzing Intrinsically Disordered Proteins in R

PONE-D-21-38609R1

Dear Dr. Yanowitz,

We’re pleased to inform you that your manuscript has been judged scientifically suitable for publication and will be formally accepted for publication once it meets all outstanding technical requirements.

Kind regards,

Eugene A. Permyakov, Ph.D., Dr.Sci.

Academic Editor

PLOS ONE
---

## [Editor Report · Acceptance letter]

8 Apr 2022

PONE-D-21-38609R1 

idpr: A package for profiling and analyzing Intrinsically Disordered Proteins in R 

Dear Dr. Yanowitz:

I'm pleased to inform you that your manuscript has been deemed suitable for publication in PLOS ONE. Congratulations! Your manuscript is now with our production department. 

Kind regards, 

on behalf of

Prof. Eugene A. Permyakov 

Academic Editor

PLOS ONE